# Numerical Modeling of Pollutant Transport: Results and Optimal Parameters

**Olaoluwa Ayodeji Jejeniwa [1], Hagos Hailu Gidey [2] and Appanah Rao Appadu [1,*]**

[1] Department of Mathematics & Applied Mathematics, Nelson Mandela University,
Gqeberha 6031, South Africa

[2] Department of Mathematics and Statistical Sciences, Botswana International University of Science
and Technology, Palapye Private Bag 16, Botswana

*   Correspondence: rao.appadu@mandela.ac.za or rao.appadu31@gmail.com

**Abstract:** In this work, we used three finite difference schemes to solve 1D and 2D convective diffusion equations. The three methods are the Kowalic–Murty scheme, Lax–Wendroff scheme, and nonstandard finite difference (NSFD) scheme. We considered a total of four numerical experiments; in all of these cases, the initial conditions consisted of symmetrical profiles. We looked at cases when the advection velocity was much greater than the diffusion of the coefficient and cases when the coefficient of diffusion was much greater than the advection velocity. The dispersion analysis of the three methods was studied for one of the cases and the optimal value of the time step size $k$, minimizing the dispersion error at a given value of the spatial step size. From our findings, we conclude that Lax–Wendroff is the most efficient scheme for all four cases. We also show that the optimal value of $k$ computed by minimizing the dispersion error at a given value of a spacial step size gave the lowest $l_2$ and $l_\infty$ errors.

**Keywords:** finite difference schemes; amplification factor; stability; relative phase error; optimization

## 1. Introduction

Petroleum is used as fuel for daily human activities. Liquid petroleum has some useful advantages over other energy sources; it is concentrated and could be easily transported from one point to another.

The major objective of oil spill modeling is to predict where oil is likely to go after a spill [1,2]. The use of data on ocean currents, winds, waves, and other environmental factors help in this regard [3]. Ovsienko et al. [4] developed a model to forecast the behavior and spreading of oil at sea using the particle-in-cell technique on a quasi-Eulerian adaptive grid. The fate and behavior of spilled oil can be affected by nine physical, chemical, and biological processes: advection, spreading, evaporation, dissolution, emulsification, dispersion, auto-oxidation, biodegradation, and sinking/sedimentation [5].

Cho et al. [1] analyzed the movement of oil with a numerical model that solved an advection–diffusion reaction equation with finite difference schemes. The spilled oil dispersion model was established in consideration of tide and tidal currents, simultaneously. They obtained the velocity components in the advection–diffusion reaction equation from the shallow water equations. Another commonly used method is the split-operator approach where the convection and diffusion terms are solved by two different numerical methods [6]. A one-dimensional convective diffusion equation was solved by Noye–Tan [7] using the third-order semi-implicit finite difference method. This approach was later extended by Noye–Tan [8] to solve the two-dimensional convective diffusion equation but the said method had issues handling three-dimensional problems because of the large matrix inversion at each time step. The quadratic upstream interpolation convective kinematics (QUICK) method for one-dimensional unsteady flow was introduced by Leonard [9] to address the issue of numerical dispersion. This method was extended to an improved

scheme, getting rid of wiggles in its entirety by introducing exponential integration into the regions with sharp fronts.

The second-order wave equation method for advection calculation (SOWMAC) was used to discretize the advection term in the advection–diffusion equation based on the characteristic method, but this method is implicit [1]. An upwind difference scheme was used by Sankaranarayanan et al. [10] for the discretization of the convective terms of the convective diffusion equation of the shallow water momentum equations. This latter scheme was constructed by Kowalik–Murty [11] and referred to as the third-order upwind scheme. However, we found the scheme to be second-order accurate in space and first-order accurate in time after conducting the truncation error analysis.

Three numerical methods have been used to solve two problems described by advection–diffusion equations in Appadu et al. [12]. The methods are third-order upwind [13], fourth-order upwind [13], and the nonstandard finite difference scheme. First-order classical methods, such as the Lax–Friedrichs method, often fail to capture shocks efficiently and odd–even decoupling usually occurs [14]. Results can be improved by employing high-order schemes. High-order schemes for convection–diffusion equations are reported in [14–19]. Use of special multi-grid strategies on non-uniform grids for solving 3D convection–diffusion problems with boundary/interior layers was reported by Ma et al. [15]. Jha and Lin [16] constructed a two-level implicit compact formulation with quasi-variable meshes for solving three-dimensional second-order non-linear parabolic partial differential equations.

## 2. Organisation of Paper

In this work, we present three numerical schemes, i.e., the Kowalic–Murty [11] scheme, Lax–Wendroff scheme, and a nonstandard finite difference scheme for solving convective diffusion equations. Section 4 introduces the one-dimensional convective diffusion equation; the three methods are introduced and their regions of stability are obtained using the von Neumann stability analysis. In Section 5, optimization analyses are carried out to determine the optimal value of the temporal step size at a given spatial step size that will minimize the dispersion errors. The graphs of the relative phase error and integrated error are displayed. In Section 6, numerical results for numerical experiment 2 for the three schemes are tabulated and the graphical representations of the solutions and behaviors of the errors are displayed. In Section 7, a 2D problem is considered and the schemes in Section 4 are extended to discretize the 2D convection–diffusion equation. Regions of stability are also obtained for the schemes. In Section 8, numerical results for the two experiments dealing with the 2D advection–diffusion equation are displayed graphically and $l_2$ and $l_\infty$ errors are tabulated. The salient features of this paper are highlighted in the Section 9.

## 3. Dispersive and Dissipative Properties

In simple room acoustic models, sound waves of different frequencies are expected to travel through the air at the same speed, but numerical dispersion causes waves of different frequencies to travel at different speeds [20]. A phenomenon of waves of different frequencies traveling at different speeds is called dispersion. It causes numerical solutions to spread out as time progresses. Dissipation is the constant decrease in the amplitude of a plane wave as time progresses. The relative phase error (RPE) is defined as the ratio of the numerical phase velocity to the exact phase velocity and is calculated as:

$$RPE = \frac{\arg(\xi_{num})}{\arg(\xi_{exact})},$$

where $\xi_{num}$ is the numerical amplification factor, which is obtained using Fourier analysis and $\xi_{exact}$ is the exact amplification factor of the partial differential equation [20].

The exact amplification factor of the UPFD scheme discretizing

$$\frac{\partial u}{\partial t}(1 + \alpha) + a\frac{\partial u}{\partial x} - D\frac{\partial^2 u}{\partial x^2} = -\kappa u,$$

was obtained in Appadu [14]. Applying the ansatz $C = \exp(\alpha t)\exp(I\theta x)$, to the PDE

$$\frac{\partial C}{\partial t} + U\frac{\partial C}{\partial x} = K_x\frac{\partial^2 C}{\partial x^2}, \tag{1}$$

we obtain the dispersion relation $\alpha = -K_x\theta^2 - IU\theta$. Since $\xi_{exact} = \frac{C(x,t+k)}{C(x,t)}$, where $C = \exp(\alpha t)\exp(I\theta x)$, [21], we obtain $\xi_{exact} = \exp\{(-K_x\theta^2 - IU\theta)k\}$. The RPE of the numerical method discretizing Equation (1) is $RPE = -\frac{1}{c_x\omega}\arg(\xi_{num})$, where $c_x = \frac{Uk}{h}$.

To compare the performance of the methods, we calculate $l_2$, $l_\infty$ errors, and the total mean square error. The $l_2$, $l_\infty$ errors, and total mean square error ($TMSE$) are calculated as [22]

$$l_2 \text{ error} = \sqrt{h\sum_{i=1}^{N}(C_i - C_i^*)^2},$$

$$l_\infty \text{ error} = \max_i |C_i - C_i^*|,$$

and [23,24]

$$TMSE = \frac{1}{N}\sum_{i=1}^{N}(C_i - C_i^*)^2,$$

where $C_i$ and $C_i^*$ are the analytical and numerical solutions, respectively, at a given spatial grid point $i$, and $N$ is the number of spatial nodes.

## 4. 1D Convection–Diffusion Equation

We consider the following equation:

$$\frac{\partial C}{\partial t} + U\frac{\partial C}{\partial x} - K_x\frac{\partial^2 C}{\partial x^2} = 0, \tag{2}$$

where $0 \leq x \leq 9, 0 \leq t \leq 5$, with initial conditions,

$$C(x,0) = \exp\left(-\frac{(x-x_0)^2}{K_x}\right),$$

and subject to the boundary condition

$$\frac{\partial C}{\partial x} = 0.$$

The exact solution is given by Sankaranarayanan et al. [10] as

$$C(x,t) = \frac{1}{\sqrt{4t+1}}\exp\left(-\frac{(x-x_0-Ut)^2}{K_x(4t+1)}\right). \tag{3}$$

We consider two cases:

(i)  numerical experiment 1, where $K_x = 0.005$, $U = 0.8$ and $x_0 = 1$, i.e., $(U >> K_x)$.
(ii) numerical experiment 2, where $K_x = 0.8$, $U = 0.005$ and $x_0 = 1$, i.e., $(U << K_x)$.

Three finite difference methods are used to solve Equation (2). We also study the stability and consistency of the methods. Stability is of major concern in the study of convergence of a numerical method for approximating a PDE, and it is often very difficult to obtain the region of stability for a numerical method. In the past, many attempts were

made to obtain stability regions of numerical methods discretizing convection–diffusion equations. In Sousa [25], several techniques from previous works have been reviewed using the forward time central space scheme. Hindmarsh et al. [26] made use of the von Neumann analysis and proved that the correct limits were necessary and sufficient stability conditions.

### 4.1. Kowalic–Murty Scheme [11]

Here, we describe how an upwind scheme was constructed by Kowalik–Murty [11] for Equation (2) in which the first derivative of $C$ with respect to $t$ and $x$ is given by

$$\frac{\partial C}{\partial t} \approx \frac{C_i^{n+1} - C_i^n}{k},$$

and

$$\frac{\partial C}{\partial x} \approx \frac{C_{i-2}^n - 6C_{i-1}^n + 3C_i^n + 2C_{i+1}^n}{6h},$$

respectively.

The second derivative of $C$ with respect to $x$ is obtained as

$$\frac{\partial^2 C}{\partial x^2} \approx \frac{C_{i+1}^n - 2C_i^n + C_{i-1}^n}{h^2},$$

where $h$ and $k$ are the spatial and temporal step sizes. Upon substituting these approximations into Equation (2), we have

$$\frac{C_i^{n+1} - C_i^n}{k} + U\left(\frac{C_{i-2}^n - 6C_{i-1}^n + 3C_i^n + 2C_{i+1}^n}{6h}\right) - K_x\left(\frac{C_{i+1}^n - 2C_i^n + C_{i-1}^n}{h^2}\right) = 0. \quad (4)$$

This can be written explicitly as

$$C_i^{n+1} = C_i^n - \frac{kU}{6h}\left(C_{i-2}^n - 6C_{i-1}^n + 3C_i^n + 2C_{i+1}^n\right) + \frac{kK_x}{h^2}\left(C_{i+1}^n - 2C_i^n + C_{i-1}^n\right), \quad (5)$$

for $i = 3, \cdots, NP - 1$, where $NP$ is the number of spatial nodes. Note that to implement this scheme, a four-point upstream formula is used near the boundary when $i = 2$ [10].

The stability region is obtained using the von Neumann stability analysis, i.e., using the ansatz $C_i^n = \xi^n e^{I\omega i}$ where $I = \sqrt{-1}$, into Equation (5) and simplifying, we obtain the amplification factor as

$$\xi = 1 - \frac{kU}{2h} - \frac{2kK_x}{h^2} + \left(\frac{2kU}{3h} + \frac{2kK_x}{h^2}\right)\cos\omega - \frac{kU}{6h}\cos 2\omega + I\left(\frac{kU}{6h}\sin 2\omega - \frac{4kU}{3h}\sin\omega\right), \quad (6)$$

where $\omega = \theta h$ is the phase angle.

For a purely non-imaginary amplification factor, we choose $\omega = \pi$ in Equation (6) and obtain

$$\xi = 1 - \frac{kU}{2h} - \frac{2kK_x}{h^2} - \frac{2kU}{3h} - \frac{2kK_x}{h^2} - \frac{kU}{6h}.$$

For stability, $|\xi| \leq 1$, and so we have the following region of stability:

$$0 < \frac{k}{h} \leq \frac{3h}{2(hU + 3K_x)}. \quad (7)$$

We next consider the case when $\omega \to 0$. When $\omega \to 0$, $\cos\omega \approx 1 - \frac{1}{2}\omega^2$, $\sin\omega \approx \omega$, and neglect higher-order terms, we have

$$|\xi|^2 = 1 - \frac{2kK_x}{h^2}\omega^2 + \frac{k^2U^2}{h^2}\omega^2. \quad (8)$$

For stability, $|\xi| \leq 1$, and we have after simplification

$$k \leq \frac{2K_x}{U^2}. \tag{9}$$

Hence, we have the region of stability for the scheme to be the intersection of the regions defined by Equations (7) and (9).

For cases 1 and 2, when $h = 0.025$, the stability regions are $0 < k \leq 1.5625 \times 10^{-2}$ and $0 < k \leq 3.9060 \times 10^{-4}$, respectively.

**Consistency of the scheme**

To determine the order of accuracy of the scheme, we apply the Taylor series expansion at about the point $(t_n, x_i)$. We then obtain,

$$\frac{\partial C}{\partial t} + U\frac{\partial C}{\partial x} - K_x\frac{\partial^2 C}{\partial x^2} = -\frac{1}{2}k\frac{\partial^2 C}{\partial t^2} - \frac{1}{6}k^2\frac{\partial^3 C}{\partial t^3} - \frac{1}{24}k^3\frac{\partial^4 C}{\partial t^4} + \frac{1}{12}K_x h^2\frac{\partial^4 C}{\partial x^4} - \frac{1}{12}Uh^3\frac{\partial^4 C}{\partial x^4} + \cdots.$$

We deduce that the scheme is of order two in space and order one in time.

*4.2. Lax–Wendroff Method*

This scheme uses the following approximations [27] for the 1D convection–diffusion equation:

$$\frac{\partial C}{\partial t} \approx \frac{C_i^{n+1} - C_i^n}{k}, \tag{10}$$

$$\frac{\partial C}{\partial x} \approx c_x\frac{C_i^n - C_{i-1}^n}{h} + (1 - c_x)\frac{C_{i+1}^n - C_{i-1}^n}{2h}, \tag{11}$$

$$\frac{\partial^2 C}{\partial x^2} \approx \frac{C_{i+1}^n - 2C_i^n + C_{i-1}^n}{h^2}, \tag{12}$$

where $c_x = \frac{Uk}{h}$. Upon substitution of Equations (10)–(12) into Equation (2), we obtain

$$\frac{C_i^{n+1} - C_i^n}{k} + Uc_x\left(\frac{C_i^n - C_{i-1}^n}{h}\right) + (1 - c_x)U\left(\frac{C_{i+1}^n - C_{i-1}^n}{2h}\right) - K_x\left(\frac{C_{i+1}^n - 2C_i^n + C_{i-1}^n}{h^2}\right) = 0, \tag{13}$$

which is written explicitly as

$$C_i^{n+1} = \frac{1}{2}(2s_x + c_x + c_x^2)C_{i-1}^n + (1 - c_x^2 - 2s_x)C_i^n + \frac{1}{2}(2s_x - c_x + c_x^2)C_{i+1}^n, \tag{14}$$

where $s_x = \frac{kK_x}{h^2}$. We now obtain the region of stability using the approach used in Hindmarsh et al. [26].

We first obtain the amplification factor, $\xi$, from the scheme, which is

$$\xi = \frac{1}{2}(2s_x + c_x + c_x^2)e^{-I\omega_x} + \frac{1}{2}(2s_x - c_x + c_x^2)e^{I\omega_x} + (1 - c_x^2 - 2s_x),$$

and then consider the case when $\omega = \pi$. The scheme is stable if the inequality $0 < |\xi| \leq 1$ holds. The Lax–Wendroff scheme is stable when it satisfies the inequality

$$2s_x + c_x^2 \leq 1. \tag{15}$$

We next consider the case when $\omega \to 0$. When $\omega \to 0$, $\cos\omega \approx 1 - \frac{1}{2}\omega^2$, $\sin\omega \approx \omega$, and neglect higher-order terms, we have

$$|\xi|^2 = 1 - 2s_x\omega^2, \tag{16}$$

this implies that $s_x > 0$. Thus, the scheme is stable when $0 \leq s_x \leq \frac{1-c_x^2}{2}$ [14].

Stability regions for cases 1 and 2 when $h = 0.025$ are $0 < k \leq 2.4400 \times 10^{-2}$ and $0 < k \leq 3.9060 \times 10^{-4}$ respectively.

**Consistency of the scheme**

To determine the order of accuracy of this scheme, we apply the Taylor series expansion at about the point $(t_n, x_i)$ on Equation (13), and on simplifying, we obtain

$$\frac{\partial C}{\partial t} + U\frac{\partial C}{\partial x} - K_x\frac{\partial^2 C}{\partial x^2} = -\frac{1}{2}k\frac{\partial^2 C}{\partial t^2} - \frac{1}{6}k^2\frac{\partial^3 C}{\partial t^3} + \frac{1}{2}kU^2\frac{\partial^2 C}{\partial x^2} - kh^2U\frac{\partial^3 C}{\partial x^3} + \frac{1}{24}K_x h^2\frac{\partial^4 C}{\partial x^4}$$
$$+ \frac{1}{24}U^2 k h^2\frac{\partial^4 C}{\partial x^4} + \cdots.$$

The scheme is accurate of order two in space and order one in time.

*4.3. Nonstandard Finite Difference Method*

Nonstandard finite difference (NSFD) methods have been widely used for the numerical approximations of differential equations [28,29]. The architect behind these classes of methods is Ronald Mickens [30]. The extension and summary of the known results up to 1994 are provided in Mickens [31]. The idea behind the construction of the NSFD schemes is that the discrete model must preserve the properties of the continuous model. Progress has been made in the theoretical understanding of the method by Anguelov and Lubuma [32].

Let

$$D_{\phi(k)}^+(u^n) = \mathfrak{F}_{\phi(k)}(f(u^n)), \tag{17}$$

at $t_n = t_0 + nk$ be the general one-step numerical scheme with the temporal step size $k$ that the approximate solution of the differential equation

$$\frac{du}{dt} = f(u), \quad u(t_0) = u_0 \geq 0, \tag{18}$$

where $D_{\phi(k)}(u^n) \approx \frac{du}{dt}$ and $\mathfrak{F}_{\phi(k)}(f(u^n))$ approximates the right-hand side of (18) and $u^n \approx u(t_n)$.

**Definition 1** ([33]). *The one-step scheme given by Equation (17) is called a nonstandard finite difference scheme if at least one of the following conditions is satisfied:*

- $$D_{\phi(k)}^+(u^n) = \frac{u^{n+1} - u^n}{\phi(k)},$$

  *where $\phi(k) = k + \mathcal{O}(k^2)$; $0 < \phi(k) < 1$,*
- *$\mathfrak{F}_{\phi(k)}(f(u^n)) = \Psi(u^n, u^{n+1}, \phi(\Delta t))$, where $\Psi(u^n, u^{n+1}, \phi(k))$ is a nonlocal approximation of the right-hand side of (18).*

**Definition 2** ([31]). *Equations (17) and (18) are said to have the same general solution if and only if*

$$u^n = u(t_n)$$

*for arbitrary $\phi(k)$.*

**Definition 3** (Exact scheme, [31]). *An exact finite difference scheme is the one for which the solution to the differential equation has the same general solution as the associated differential equation.*

Here, we describe how NSFD was constructed by Mickens [28] for the 1D convection–diffusion equation.

The equation

$$\frac{\partial C}{\partial t} + U\frac{\partial C}{\partial x} - K_x\frac{\partial^2 C}{\partial x^2} = 0$$

is split into three sub-equations following [34]:

$$\frac{\partial C}{\partial t} + U\frac{\partial C}{\partial x} = 0, \tag{19}$$

$$\frac{\partial C}{\partial x} = \frac{K_x}{U}\frac{\partial^2 C}{\partial x^2}, \tag{20}$$

and

$$\frac{\partial C}{\partial t} = K_x\frac{\partial^2 C}{\partial x^2}. \tag{21}$$

Equations (19) and (20) have known–exact finite difference schemes, which are

$$\frac{C_i^{n+1} - C_i^n}{k} + U\left(\frac{C_i^n - C_{i-1}^n}{h}\right) = 0,$$

and

$$U\left(\frac{C_i - C_{i-1}}{h}\right) = U\frac{C_{i+1} - 2C_i + C_{i-1}}{h(\exp(Uh/K_x) - 1)},$$

respectively, when $h \to 0$ and $k \to 0$.

The NSFD is given by Mickens [28,34] as:

$$\frac{C_i^{n+1} - C_i^n}{k} + U\left(\frac{C_i^n - C_{i-1}^n}{h}\right) = U\frac{C_{i+1}^n - 2C_i^n + C_{i-1}^n}{h(\exp(Uh/K_x) - 1)}, \tag{22}$$

which is written explicitly as

$$C_i^{n+1} = (c_x + \beta_x)\,C_{i-1}^n + (1 - c_x - 2\beta_x)\,C_i^n + \beta_x\,C_{i+1}^n, \tag{23}$$

where $c_x = \frac{kU}{h}$ and $\beta_x = \frac{Uk}{h(\exp(Uh/K_x)-1)}$. The square of the modulus of the amplification factor is given by

$$|\xi|^2 = ((1 - c_x - 2\beta_x) + (c_x + 2\,\beta_x)\cos(\omega))^2 + (c_x\sin(\omega))^2. \tag{24}$$

For stability, $0 < |\xi| \le 1$ and this implies that $0 < |\xi|^2 \le 1$. The square of the modulus of the amplification factor when $\omega = \pi$ is given by

$$|\xi|^2 = (1 - 2\,c_x - 4\,\beta_x)^2. \tag{25}$$

and, therefore,

$$c_x + 2\,\beta_x \le 1. \tag{26}$$

We next consider the case when $\omega \to 0$. When $\omega \to 0$, we use the approximations $\cos(\omega) \approx 1 - \frac{1}{2}\omega^2$, $\sin(\omega) \approx \omega$ and neglecting higher-order terms, we have

$$|\xi|^2 = 1 - (c_x + 2\,\beta_x - c_x^2)\omega^2. \tag{27}$$

$|\xi|^2 \le 1$ implies that

$$c_x + 2\,\beta_x - c_x^2 \ge 0. \tag{28}$$

Thus, the scheme is stable if it satisfies the inequality [12]

$$c_x^2 \leq c_x + 2\,\beta_x \leq 1. \tag{29}$$

Stability regions for NSFD schemes discretizing Equation (2) for cases 1 and 2 when $h = 0.025$ are $0 < k \leq 3.0700 \times 10^{-2}$ and $0 < k \leq 3.9676 \times 10^{-4}$, respectively.

**Consistency of the scheme**

To determine the order of accuracy of this scheme, we apply the Taylor series expansion at about the point $(t_n, x_i)$ in Equation (22), and upon simplifying, we obtain

$$\frac{\partial C}{\partial t} + U\frac{\partial C}{\partial x} - K_x\frac{\partial^2 C}{\partial x^2} = \frac{1}{2}hU\frac{\partial^2 C}{\partial x^2} - \frac{1}{2}k\frac{\partial^2 C}{\partial t^2} - \frac{1}{6}k^2\frac{\partial^3 C}{\partial t^3} - \frac{1}{6}Uh^2\frac{\partial^3 C}{\partial x^3} + \cdots$$

This scheme is accurate in order one in both space and time.

## 5. Optimization and Results for Numerical Experiment 1

While running numerical experiment 1 using Lax–Wendroff and NSFD schemes with $h = 0.025$, and different values of $k$ for which methods are stable, we observed that the results are much affected by the values of $k$ used especially with regard to the phase lag/lead behavior. Therefore, we decided to compute the optimal value of $k$, which minimizes the dispersion error when $h = 0.025$ for the two methods when $U = 0.8$ and $K_x = 0.005$.

We follow the same ideas as in the work of Appadu [35,36] to compute the optimal value of $k$ at a given value of $h$ by minimizing the dispersion error.

The 3D plots of the exact RPE versus $k$ vs. $\omega \in [0, 1.1]$ for the three methods: Kowalic–Murty [11], Lax–Wendroff, and NSFD are shown in Figure 1a–c. We observe that there is no phase-wrapping phenomenon [37].

The integrated error from Tam and Webb (IETAM) is defined as [35,36]

$$IETAM = \int_0^{1.1} |1 - RPE|^2 \, d\omega.$$

Plots of the integrated error versus $k$ are depicted in Figure 2. The optimal values are obtained using NLPSolve maple solver and are given as follows:

- Lax–Wendroff, optimal $k = 0.015625$;
- NSFD, optimal $k = 0.014952$.

Figure 3 shows the plots of the relative phase errors versus phase angles for different values of $k$ for all schemes. Figure 3b,c show that the RPE is closest to 1 in the case of Lax–Wendroff and NSFD methods when $k = 0.015625$ and $k = 0.014952$, respectively.

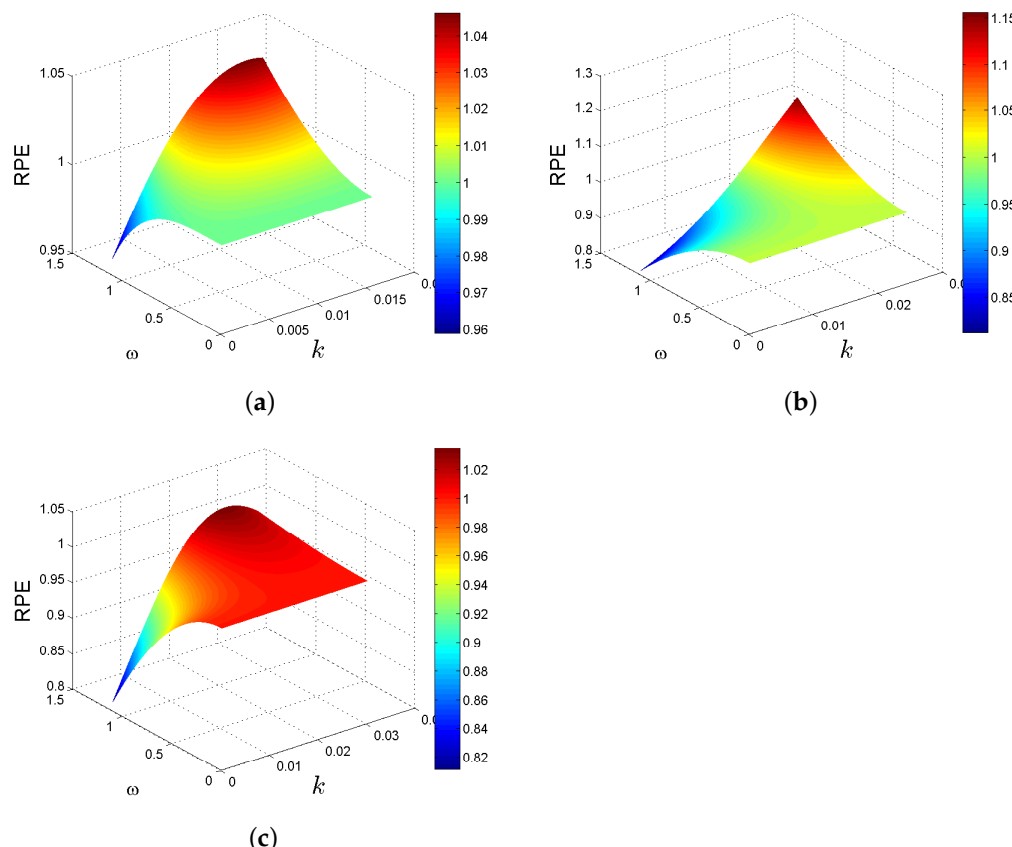

**Figure 1.** Plot of the exact RPE vs. $k$ vs. $\omega \in [0, 1.1]$ for the three methods using $h = 0.025$. (**a**) Kowalic–Murty with $k \in [0, 1.5625 \times 10^{-2}]$; (**b**) Lax–Wendroff with $k \in [0, 2.44 \times 10^{-2}]$; (**c**) NSFD with $k \in [0, 3.0700 \times 10^{-2}]$.

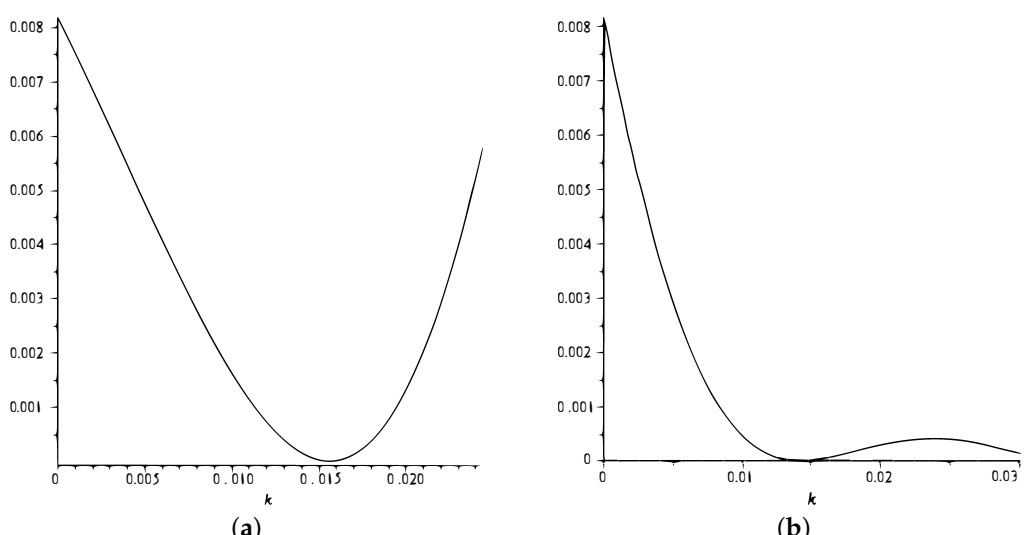

**Figure 2.** Plot of integrated error vs. $k$ to determine optimal $k$ when $h = 0.025$. (**a**) Lax–Wendroff scheme; (**b**) NSFD scheme.

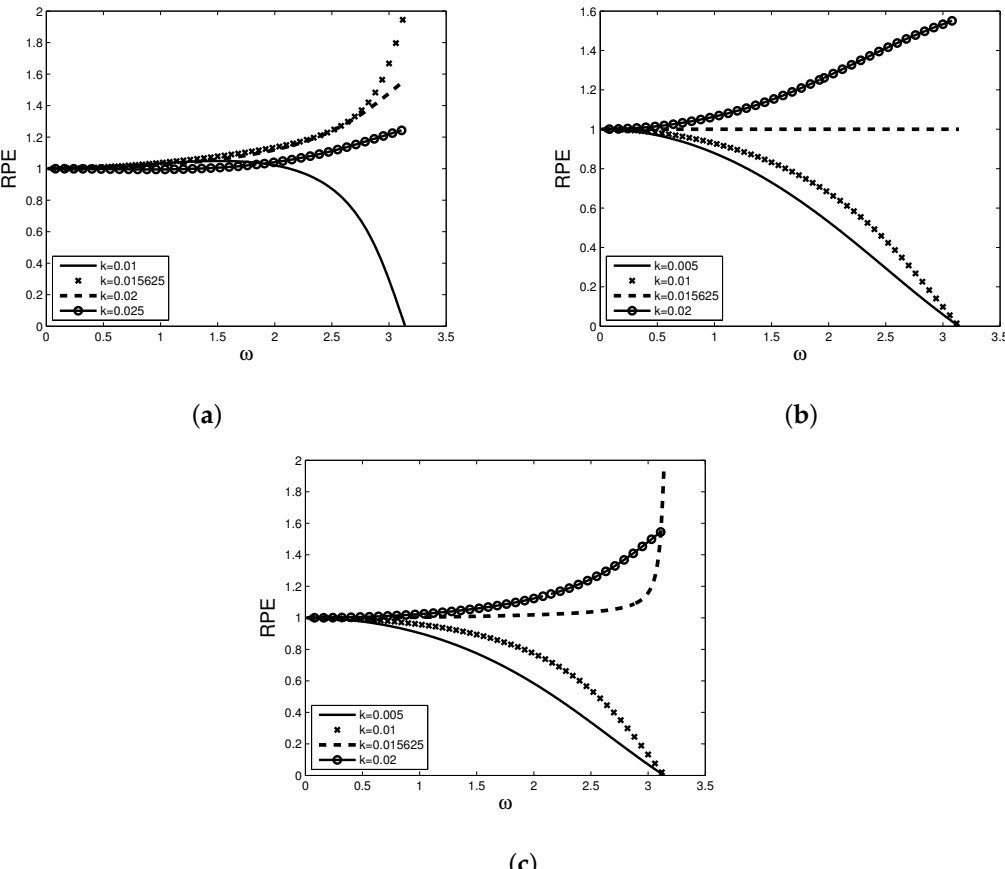

**Figure 3.** Plot of the exact RPE vs. $\omega \in [0, \pi]$ for the three methods at different values of $k$ when $h = 0.025$. (**a**) Kowalic–Murty. (**b**) Lax–Wendroff. (**c**) NSFD.

We note that for numerical experiment 1 at $h = 0.025$, the stability regions for the Kowalic–Murty scheme [11], Lax–Wendroff scheme, and NSFD scheme are $0 < k \leq 1.5625 \times 10^{-2}$, $0 < k \leq 2.4400 \times 10^{-2}$ and $0 < k \leq 3.0700 \times 10^{-2}$ respectively.

Figure 4a shows the plot of the initial and exact profiles for the transport of the one-dimensional Gaussian pulse while Figure 4b–d show the plots of the numerical and exact solutions of transport of one-dimensional Gaussian pulse of unit height, when the coefficient of convection is greater than the coefficient of diffusion after 5 s at different temporal step sizes, using the three schemes. It is observed that the pulse of the height unit decreases after 5 s. In Figure 4b, the Kowalic–Murty scheme [11] shows anti-diffusion whereby the peak of the numerical profile is higher than the peak of the exact profiles at some values of $k$. Moreover, $l_2$ and $l_\infty$ errors, as well as the total mean square error, are tabulated in Table 1 while the numerical $l_2$ rate of convergence in space is given in Table 2 for Kowalic–Murty [11], Lax–Wendroff, and NSFD schemes. For experiment 1, to obtain the rate of convergence, for a given $h$, the size of $k$ is chosen so that it lies in the region of stability. For $h = 0.1$, we use $k = 0.025$ and dividing $k$ by 2 as $h$ decreases by half.

**Table 1.** $l_2$, $l_\infty$ errors, and total mean square error at time $T = 5$ with different values of $k$ using Kowalic–Murty, Lax–Wendroff, and NSFD schemes using $h = 0.025$ for numerical experiment 1.

| Schemes | $k$ | $l_2$ Error | $l_\infty$ Error | TMSE |
|---|---|---|---|---|
| Kowalic–Murty | 0.0125 | $9.4300 \times 10^{-2}$ | $2.0600 \times 10^{-1}$ | $9.8620 \times 10^{-4}$ |
| | 0.0050 | $2.2100 \times 10^{-2}$ | $4.1300 \times 10^{-2}$ | $5.3910 \times 10^{-5}$ |
| | 0.0025 | $9.6000 \times 10^{-3}$ | $1.7400 \times 10^{-2}$ | $1.0223 \times 10^{-5}$ |
| | 0.00125 | $4.3000 \times 10^{-3}$ | $7.7000 \times 10^{-3}$ | $2.0950 \times 10^{-6}$ |
| Lax–Wendroff | 0.0200 | $2.4000 \times 10^{-3}$ | $3.9000 \times 10^{-3}$ | $6.6135 \times 10^{-7}$ |
| | **0.015625** | $\mathbf{8.4219 \times 10^{-5}}$ | $\mathbf{1.5464 \times 10^{-4}}$ | $\mathbf{7.8592 \times 10^{-10}}$ |
| | 0.0100 | $2.8000 \times 10^{-3}$ | $4.4000 \times 10^{-3}$ | $8.3834 \times 10^{-7}$ |
| | 0.0050 | $4.8000 \times 10^{-3}$ | $7.8000 \times 10^{-3}$ | $2.5858 \times 10^{-6}$ |
| NSFD | 0.0200 | $1.6600 \times 10^{-2}$ | $3.1000 \times 10^{-2}$ | $3.0480 \times 10^{-5}$ |
| | **0.0149518** | $\mathbf{4.7000 \times 10^{-3}}$ | $\mathbf{8.0000 \times 10^{-3}}$ | $\mathbf{2.4253 \times 10^{-6}}$ |
| | 0.0100 | $1.8500 \times 10^{-2}$ | $3.2200 \times 10^{-2}$ | $3.7738 \times 10^{-5}$ |
| | 0.0050 | $2.9200 \times 10^{-2}$ | $4.9900 \times 10^{-2}$ | $9.4241 \times 10^{-5}$ |

**Table 2.** Rate of convergence in space for the three schemes when used to solve numerical experiment 1 at time $T = 5$.

| Schemes | $h$ | $l_2$ Error | $l_\infty$ Error | Rate of Convergence in Space ($l_2$) |
|---|---|---|---|---|
| Kowalic–Murty | 0.0500 | 0.0471 | 0.0865 | – |
| | 0.0250 | 0.0096 | 0.0174 | 2.2946 |
| | 0.0125 | 0.0023 | 0.0042 | 2.0614 |
| | 0.00625 | $5.780 \times 10^{-4}$ | 0,0010 | 1.9925 |
| Lax–Wendroff | 0.1000 | 0.0609 | 0.0868 | – |
| | 0.0500 | 0.0221 | 0.0333 | 1.5992 |
| | 0.0250 | 0.0043 | 0.0070 | 2.2248 |
| | 0.0125 | $5.9417 \times 10^{-4}$ | $9.4389 \times 10^{-4}$ | 2.8554 |
| | 0.00625 | $9.8965 \times 10^{-5}$ | $1.5663 \times 10^{-5}$ | 2.5859 |
| NSFD | 0.1000 | 0.0814 | 0.1290 | – |
| | 0.0500 | 0.0576 | 0.0943 | 0.4990 |
| | 0.0250 | 0.0267 | 0.0460 | 1.1092 |

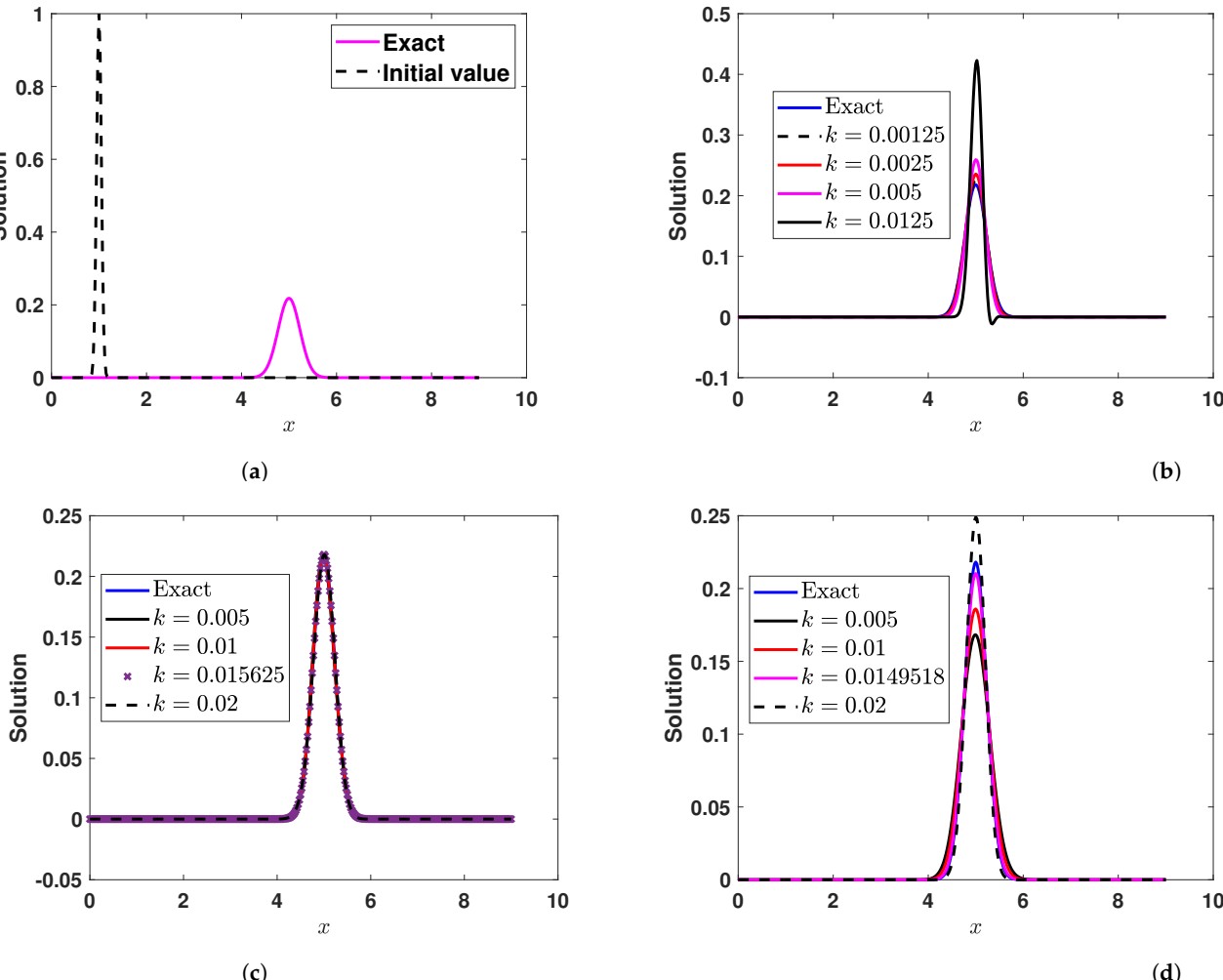

**Figure 4.** Plots of numerical and exact solution vs. $x$ at time $T = 5$ when $K_x = 0.005$ m$^2$/s and $U = 0.8$ m/s using the three schemes at $h = 0.025$ and different values of $k$. (**a**) Initial and Exact; (**b**) Kowalic–Murty; (**c**) Lax–Wendroff; (**d**) NSFD.

In Figure 5, we obtain plots of absolute error versus $x$ for each of the three schemes for two scenarios; when the scheme performs best and when it performs worst, when $h = 0.025$.

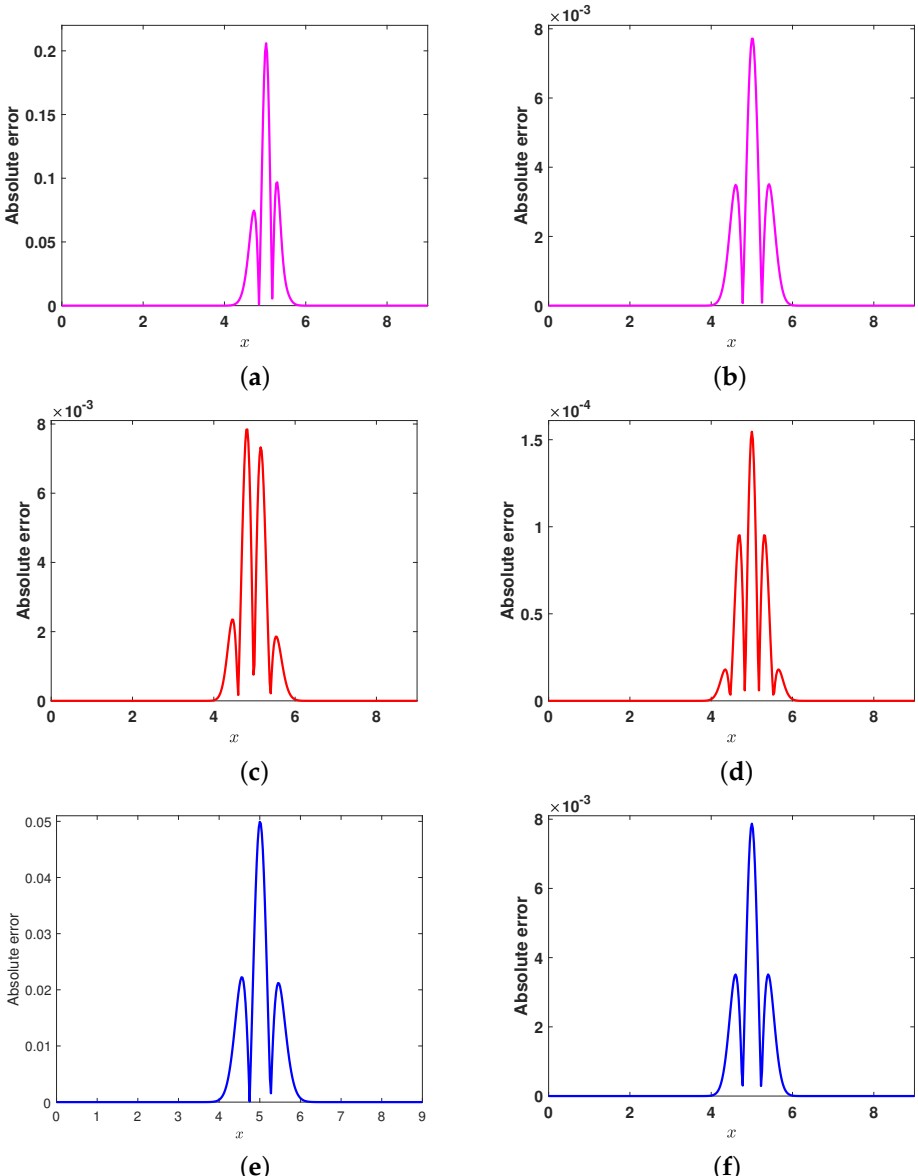

**Figure 5.** Plot of absolute errors vs. $x$ using the three methods at $h = 0.025$ and different values of $k$. (**a**) Kowalic–Murty scheme with $k = 0.0125$; (**b**) Kowalic–Murty scheme with $k = 0.00125$; (**c**) Lax–Wendroff scheme with $k = 0.0050$; (**d**) Lax–Wendroff scheme with $k = 0.015625$; (**e**) NSFD scheme with $k = 0.0050$; (**f**) NSFD scheme with $k = 0.0149518$.

## 6. Results of Numerical Experiment 2

In this section, plots of the numerical solution for the transport of the one-dimensional Gaussian pulse for numerical experiment 2 at different values of $k$ are displayed alongside the tables showing $l_2$, $l_\infty$ errors, and the rate of convergence for the three schemes considered. Figure 6 shows the plots of the initial, numerical, and exact solutions of transport of the one-dimensional Gaussian pulse of unit height using the three schemes: Kowalic–Murty [11], Lax–Wendroff, and NSFD schemes, respectively, where the coefficient of diffusion is greater than that of convection. A drastic change in the initial pulse is observed as time progresses as depicted in Figure 6. Table 3 shows the $l_2$, $l_\infty$ errors, and rate of convergence for the three schemes for numerical experiment 2. For experiment 2, to obtain the rate of convergence, for a given $h$, the size of $k$ is chosen so that it lies in the region of stability. For $h = 0.1$, we use $k = 0.005$ and divide $k$ by 4 as $h$ decreases by half. We observe that all three methods are efficient at solving numerical experiment 2; the most

efficient method here is Lax–Wendroff followed by Kowalic–Murty scheme [11], followed by the NSFD scheme.

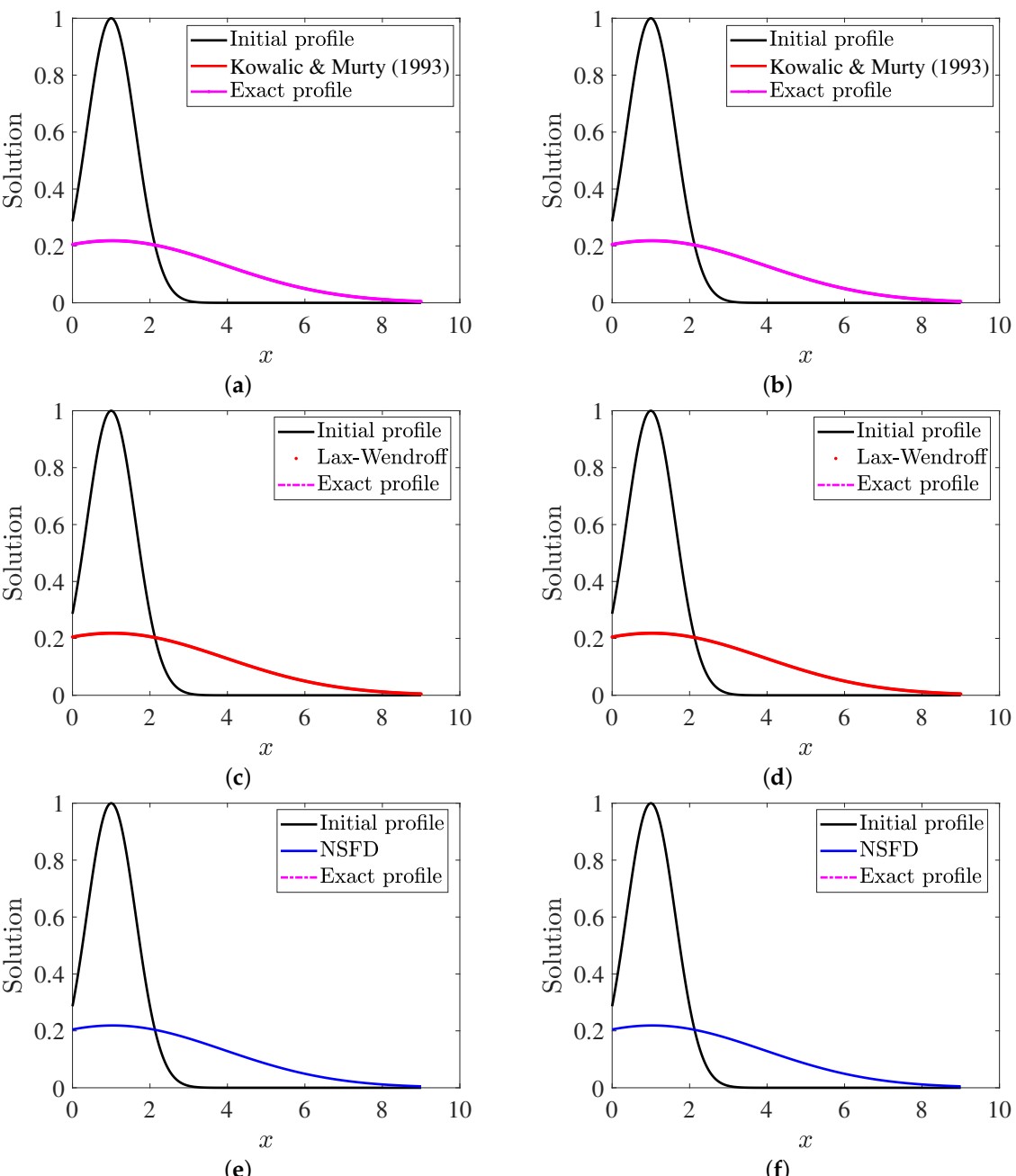

**Figure 6.** Plots of the initial profile, numerical, and exact solutions at time $T = 5$ and $h = 0.025$ when $K_x = 0.8 \text{ m}^2/\text{s}$ and $U = 0.005 \text{ m/s}$ using the three schemes at different values of $k$. (**a**) $k = 3.5 \times 10^{-4}$; (**b**) $k = 2.0 \times 10^{-4}$; (**c**) $k = 3.5 \times 10^{-4}$; (**d**) $k = 2.0 \times 10^{-4}$; (**e**) $k = 3.5 \times 10^{-4}$; (**f**) $k = 2.0 \times 10^{-4}$.

**Table 3.** Rate of convergence in space using the three schemes for numerical experiment 2 at time $T = 5$.

| Schemes | $h$ | $l_2$ Error | $l_\infty$ Error | Rate of Convergence in Space ($l_2$) |
|---|---|---|---|---|
| Kowalic–Murty scheme [11] | 0.1000 | $4.6638 \times 10^{-5}$ | $2.4520 \times 10^{-5}$ | – |
| | 0.0500 | $1.1659 \times 10^{-5}$ | $6.1359 \times 10^{-6}$ | 2.0001 |
| | 0.0250 | $2.9146 \times 10^{-6}$ | $1.5351 \times 10^{-6}$ | 2.0001 |
| | 0.0125 | $7.2862 \times 10^{-7}$ | $3.8379 \times 10^{-7}$ | 2.0001 |
| Lax–Wendroff | 0.1000 | $4.6750 \times 10^{-5}$ | $2.4172 \times 10^{-5}$ | – |
| | 0.0500 | $1.1689 \times 10^{-5}$ | $6.0506 \times 10^{-6}$ | 1.9998 |
| | 0.0250 | $2.9225 \times 10^{-6}$ | $1.5132 \times 10^{-6}$ | 1.9999 |
| | 0.0125 | $7.3062 \times 10^{-7}$ | $3.7833 \times 10^{-7}$ | 2.0000 |
| NSFD | 0.1000 | 0.0065 | 0.0032 | – |
| | 0.0500 | 0.0032 | 0.0016 | 1.0224 |
| | 0.0250 | 0.0016 | $7.8017 \times 10^{-4}$ | 1.0000 |
| | 0.0125 | $7.9874 \times 10^{-4}$ | $3.8784 \times 10^{-4}$ | 1.0023 |

## 7. 2D Advection–Diffusion Equation

We solve

$$\frac{\partial C}{\partial t} + U\frac{\partial C}{\partial x} + V\frac{\partial C}{\partial y} - K_x\frac{\partial^2 C}{\partial x^2} - K_y\frac{\partial^2 C}{\partial y^2} = 0, \tag{30}$$

subject to the boundary condition

$$\frac{\partial C}{\partial x} = \frac{\partial C}{\partial y} = 0,$$

and domains being $0 \le x, y \le a,\ 0 < t \le c$.

**Numerical experiment 3**

Here, the initial condition $C(x, y, t_0)$ is selected from the exact solution given by

$$C(x, y, t) = \frac{1}{4t+1} \exp\left(-\frac{(x - x_0 - Ut)^2}{K_x(4t+1)} - \frac{(y - y_0 - Vt)^2}{K_y(4t+1)}\right), \tag{31}$$

where $0 \le x, y \le 6, 0 < t \le 5$, $(x_0, y_0) = (0.5, 0.5)$, $U = V = 0.8$ and $K_x = K_y = 0.01$.

**Numerical experiment 4**

Here, the initial condition $C(x, y, t_0)$ is selected from the exact solution given by

$$C(x, y, t) = \frac{K}{\sqrt{4\pi K_x t}\sqrt{4\pi K_y t}} \exp\left(-\frac{(x - x_0 - Ut)^2}{4K_x t} - \frac{(y - y_0 - Vt)^2}{4K_y t}\right), \tag{32}$$

where $0 \le x, y \le 100,000, 0 < t \le 36,000$, $(x_0, y_0) = (50,000, 50,000)$, $U = V = 0.5$ and $K_x = K_y = 10,000$, $K = 10^{12}$.

### 7.1. 2D Scheme from Kowalic–Murty Scheme

Here, we use the second-order upwind scheme for the 2D convection–diffusion Equation (30) given by

$$\frac{C_{i,j}^{n+1} - C_{i,j}^{n}}{k} + U\left(\frac{C_{i-2,j}^{n} - 6C_{i-1,j}^{n} + 3C_{i,j}^{n} + 2C_{i+1,j}^{n}}{6h}\right) + V\left(\frac{C_{i,j-2}^{n} - 6C_{i,j-1}^{n} + 3C_{i,j}^{n} + 2C_{i,j+1}^{n}}{6h}\right)$$

$$- K_x\left(\frac{C_{i+1,j}^{n} - 2C_{i,j}^{n} + C_{i-1,j}^{n}}{h_x^2}\right) - K_y\left(\frac{C_{i,j+1}^{n} - 2C_{i,j}^{n} + C_{i,j-1}^{n}}{h_y^2}\right) = 0. \tag{33}$$

This can be written explicitly as

$$C_{i,j}^{n+1} = C_{i,j}^{n} - \frac{kU}{6h_x}\{C_{i-2,j}^{n} - 6C_{i-1,j}^{n} + 3C_{i,j}^{n} + 2C_{i+1,j}^{n}\} -$$

$$\frac{kV}{6h_y}\{C_{i,j-2}^{n} - 6C_{i,j-1}^{n} + 3C_{i,j}^{n} + 2C_{i,j+1}^{n}\} + \tag{34}$$

$$\frac{kK_x}{h_x^2}\{C_{i+1,j}^{n} - 2C_{i,j}^{n} + C_{i-1,j}^{n}\} + \frac{kK_y}{h_y^2}\{C_{i,j+1}^{n} - 2C_{i,j}^{n} + C_{i,j-1}^{n}\}.$$

The stability region is obtained using the von Neumann stability analysis, i.e., substituting the ansatz $C_{i,j}^{n} = \xi^n e^{I(\omega_x i + \omega_y j)}$, where $I = \sqrt{-1}$ into Equation (34), and simplifying, we obtain the amplification factor

$$\xi = 1 - \frac{kU}{2h_x} - \frac{2kK_x}{h_x^2} - \frac{kV}{2h_y} - \frac{2kK_y}{h_y^2} + \left(\frac{2kU}{3h_x} + \frac{2kK_x}{h_x^2}\right)\cos\omega_x +$$

$$\left(\frac{2kV}{3h_x} + \frac{2kK_y}{h_y^2}\right)\cos\omega_y - \frac{kU}{6h_x}\cos 2\omega_x - \frac{kV}{6h_y}\cos 2\omega_y + \tag{35}$$

$$I\left\{\frac{kU}{6h_x}\sin 2\omega_x - \frac{4kU}{3h_x}\sin\omega_x + \frac{kV}{6h_y}\sin 2\omega_y - \frac{4kV}{3h_y}\sin\omega_y\right\}.$$

For the purely non-imaginary amplification factor, we choose $\omega_x = \omega_y = \pi$. This gives

$$\xi = 1 - \frac{4kU}{3h_x} - \frac{4kK_x}{h_x^2} - \frac{4kV}{3h_y} - \frac{4kK_y}{h_y^2}.$$

For stability, $|\xi| \leq 1$, and so we have the following region of stability

$$0 < \frac{k}{h} \leq \frac{3h}{2\{(U+V) + 3(K_x + K_y)\}}, \tag{36}$$

where we have chosen $h_x = h_y = h$.

For $\omega_x \to 0$ and $\omega_y \to 0$, we use Taylor's approximation; $\cos\omega_x \approx 1 - \omega_x^2/2$, $\sin\omega_x \approx \omega_x$, and on neglecting the higher-order terms, we have the following condition

$$\xi^2 = 1 - \frac{2kK_x}{h_x^2}\omega^2 + \frac{k^2 U^2}{h_x^2}\omega^2 - \frac{2kK_y}{h_y^2}\omega^2 + \frac{k^2 V^2}{h_y^2}\omega^2. \tag{37}$$

For stability, $|\xi| \leq 1$, after simplification, we have

$$k \leq \frac{2(K_x + K_y)}{U^2 + V^2}, \tag{38}$$

here, we chose $h_x = h_y = h$. Hence, we have the region of stability for the scheme to be the intersection of the regions defined by (36) and (38).

### 7.2. 2D Lax–Wendroff Scheme

This scheme uses the following approximations for Equation (30)

$$
\frac{\partial C}{\partial t} \approx \frac{C_{i,j}^{n+1} - C_{i,j}^n}{k},
$$
$$
\frac{\partial C}{\partial x} \approx c_x \left( \frac{C_{i,j}^n - C_{i-1,j}^n}{h_x} \right) + (1 - c_x) \left( \frac{C_{i+1,j}^n - C_{i-1,j}^n}{2h_x} \right),
$$
$$
\frac{\partial C}{\partial y} \approx c_y \left( \frac{C_{i,j}^n - C_{i,j-1}^n}{h_y} \right) + (1 - c_y) \left( \frac{C_{i,j+1}^n - C_{i,j-1}^n}{2h_y} \right),
$$
$$
\frac{\partial^2 C}{\partial x^2} \approx \frac{C_{i+1,j}^n - 2C_{i,j}^n - C_{i-1,j}^n}{h_x^2},
$$
$$
\frac{\partial^2 C}{\partial y^2} \approx \frac{C_{i,j+1}^n - 2C_{i,j}^n - C_{i,j-1}^n}{h_y^2},
$$

where $c_x = \frac{Uk}{h_x}$ and $c_y = \frac{Vk}{h_y}$.

Substituting these approximations into Equation (30), we obtain

$$
\frac{C_{i,j}^{n+1} - C_{i,j}^n}{k} + Uc_x \left( \frac{C_{i,j}^n - C_{i-1,j}^n}{h_x} \right) + U(1 - c_x) \left( \frac{C_{i+1,j}^n - C_{i-1,j}^n}{2h_x} \right) + Vc_y \left( \frac{C_{i,j}^n - C_{i,j-1}^n}{h_y} \right) +
$$
$$
V(1 - c_y) \left( \frac{C_{i,j+1}^n - C_{i,j-1}^n}{2h_y} \right) - k_x \left( \frac{C_{i+1,j}^n - 2C_{i,j}^n - C_{i-1,j}^n}{h_x^2} \right) - K_y \left( \frac{C_{i,j+1}^n - 2C_{i,j}^n - C_{i,j-1}^n}{h_y^2} \right) = 0. \tag{40}
$$

This is written explicitly as

$$
C_{i,j}^{n+1} = \frac{1}{2}(2s_x + c_x + c_x^2) C_{i-1,j}^n + \frac{1}{2}(2s_y + c_y + c_y^2) C_{i,j-1}^n + (1 - c_x^2 - 2s_x - c_y^2 - 2s_y) C_{i,j}^n
$$
$$
+ \frac{1}{2}(2s_x - c_x + c_x^2) C_{i+1,j}^n + \frac{1}{2}(2s_y - c_y + c_y^2) C_{i,j+1}^n, \tag{41}
$$

where $s_x = \frac{kK_x}{h_x^2}$ and $s_y = \frac{kK_y}{h_y^2}$.

For stability, we use the approach of Hindmarsh et al. [26]. Using the von Neumann stability, the amplification factor is given by

$$
\xi = \frac{1}{2}(2s_x + c_x + c_x^2) e^{-I\omega_x} + \frac{1}{2}(2s_y + c_y + c_y^2) e^{-I\omega_y} + (1 - c_x^2 - 2s_x - c_y^2 - 2s_y)
$$
$$
+ \frac{1}{2}(2s_x - c_x + c_x^2) e^{I\omega_x} + \frac{1}{2}(2s_y - c_y + c_y^2) e^{I\omega_y}.
$$

For $\omega_x = \omega_y = \pi$, $|\xi| \leq 1$, gives the inequality

$$
2(s_x + s_y) + c_x^2 + c_y^2 \leq 1. \tag{42}
$$

For $\omega_x \to 0$ and $\omega_y \to 0$, we use Taylor's approximation, and upon neglecting the higher-order terms, we have the following condition

$$
c_x c_y \leq s_x + s_y. \tag{43}
$$

Thus, the scheme is stable when

$$
c_x c_y \leq s_x + s_y \leq \frac{1}{2}(1 - c_x^2 - c_y^2).
$$

*7.3. 2D NSFD Scheme*

The NSFD scheme for the 2D convection–diffusion equation given by Equation (30) is

$$\frac{C_{i,j}^{n+1} - C_{i,j}^n}{k} + U\left(\frac{C_{i,j}^n - C_{i-1,j}^n}{h_x}\right) + V\left(\frac{C_{i,j}^n - C_{i,j-1}^n}{h_y}\right) =$$

$$U\left(\frac{C_{i+1,j}^n - 2C_{i,j}^n + C_{i-1,j}^n}{h_x(\exp(h_x U / K_x) - 1)}\right) + V\left(\frac{C_{i,j+1}^n - 2C_{i,j}^n + C_{i,j-1}^n}{h_y(\exp(h_y V / K_y) - 1)}\right), \qquad (44)$$

which is written explicitly as

$$C_{i,j}^{n+1} = (c_x + \beta_x)\, C_{i-1,j}^n + (c_y + \beta_y)\, C_{i,j-1}^n + (1 - c_x - 2\beta_x - c_y - 2\beta_y)\, C_{i,j}^n$$

$$+ \beta_x\, C_{i+1,j}^n + \beta_y\, C_{i,j+1}^n, \qquad (45)$$

where $c_x = \frac{Uk}{h_x}$, $c_y = \frac{Vk}{h_y}$, $\beta_x = \dfrac{kU}{h_x \exp(h_x U / K_x) - 1}$ and $\beta_y = \dfrac{kV}{h_y \exp(h_y V / K_y) - 1}$.

Using von Neumann stability, the amplification factor is given by

$$\xi = (c_x + \beta_x)\, e^{-I\omega_x} + (c_y + \beta_y)\, e^{-I\omega_y} + (1 - c_x - 2\beta_x - c_y - 2\beta_y) + \beta_x\, e^{I\omega_x} + \beta_y\, e^{-I\omega_y}$$

For $\omega_x = \omega_y = \pi$, we have

$$\xi = -(c_x + \beta_x) - (c_y + \beta_y) + (1 - c_x - 2\beta_x - c_y - 2\beta_y) - \beta_x - \beta_y.$$

The scheme is stable when the following inequality is satisfied

$$c_x + c_y + 2\,(\beta_x + \beta_y) \leq 1. \qquad (46)$$

For $\omega_x \to 0$ and $\omega_y \to 0$, we use Taylor's approximation, and upon neglecting the higher-order terms, for stability, we have the following condition

$$(c_x + c_y)^2 \leq c_x + c_y + 2\,(\beta_x + \beta_y). \qquad (47)$$

Thus, the scheme is stable when

$$(c_x + c_y)^2 \leq c_x + c_y + 2\,(\beta_x + \beta_y) \leq 1.$$

Table 4 presents the range of values of $k$ for the stability of the three methods for numerical experiments 3 and 4 at selected values of $h$.

**Table 4.** Stability region of the three methods discretizing 2D advection–diffusion equation for numerical experiments 3 and 4.

| Schemes | Numerical Experiment | Value of $h$ | Stability Region |
|---|---|---|---|
| Kowalic–Murty | 3 | 0.025 | $0 < k \leq 9.3750 \times 10^{-3}$ |
| | 4 | 5000 | $0 < k \leq 624.99$ |
| Lax–Wendroff | 3 | 0.025 | $0 < k \leq 0.0114$ |
| | 4 | 5000 | $0 < k \leq 624.99$ |
| NSFD | 3 | 0.025 | $0 < k \leq 0.0128$ |
| | 4 | 5000 | $0 < k \leq 697.74$ |

## 8. Some Numerical Results

### 8.1. Numerical Experiment 3

In this section, we present the numerical solutions of the two-dimensional convection–diffusion equation for the case when $h_x = h_y = 0.025$, $U = V = 0.8$, $K_x = K_y = 0.01$. The plots of the initial solution, the numerical and exact solutions, as well as the $l_2$, $l_\infty$, and relative errors, are displayed and tabulated for the three schemes considered. In this case, the initial Gaussian pulse of the unit height located at $(x_0, y_0) = (0.5, 0.5)$ has moved to a new location with a decreased height at time $T = 5$. Figure 7 shows the 3D plot of the initial and exact solutions at time $T = 5$ vs. $x$ vs. $y$.

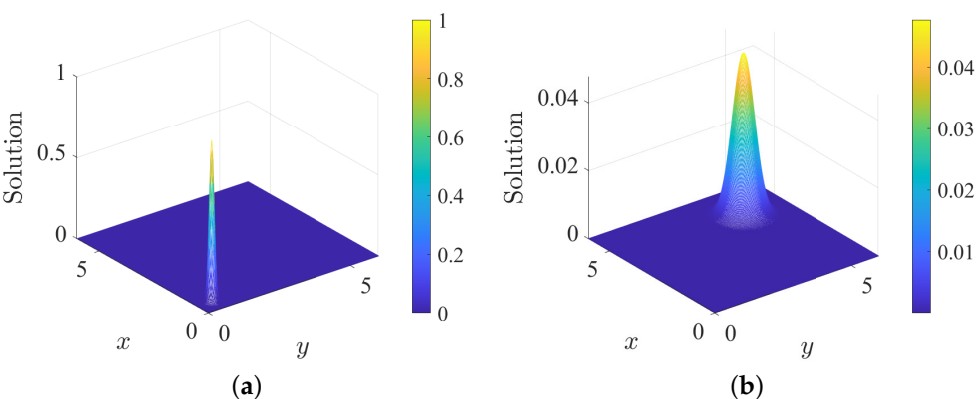

(**a**)                    (**b**)

**Figure 7.** The 3D plot of solution vs. $x$ vs. $y$ at time $T = 5$ with $U = V = 0.8$, $K_x = K_y = 0.01$, $x, y \in [0, 6] \times [0, 6]$. (**a**) Initial profile; (**b**) exact profile.

Plots of the numerical solution versus $x$ versus $y$ at time $T = 5$ for numerical experiment 3 are displayed in Figures 8–10 using the Kowalic–Murty scheme [11], Lax–Wendroff scheme, and NSFD scheme. The numerical solution agrees with the analytical as the temporal step size decreases. We also obtain $l_2$, $l_\infty$ errors in Table 5 along with the numerical rate of convergence in time and we observe that this rate of convergence is close to the theoretical rate of convergence. Using Table 5, we can also conclude that, in general, Lax–Wendroff is the most efficient, followed by NSFD, followed by the Kowalic–Murty scheme [11], as far as numerical experiment 3 is concerned.

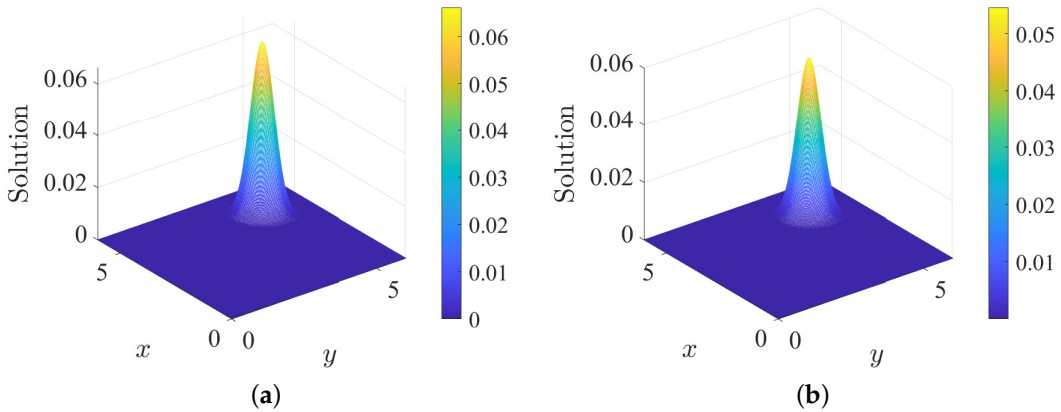

(**a**)                    (**b**)

**Figure 8.** *Cont.*

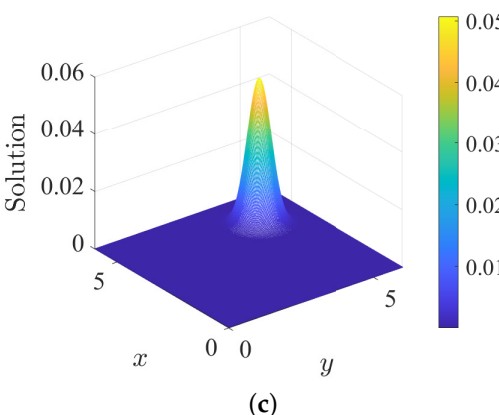

(c)

**Figure 8.** The 3D plot of the solution using the Kowalic–Murty scheme vs. $x$ vs. $y$ at time $T = 5$ with $h = 0.025$, $U = V = 0.8$, $K_x = K_y = 0.01$, $x, y \in [0, 6] \times [0, 6]$. (**a**) $k = 0.00625$; (**b**) $k = 0.0040$; (**c**) $k = 0.0025$.

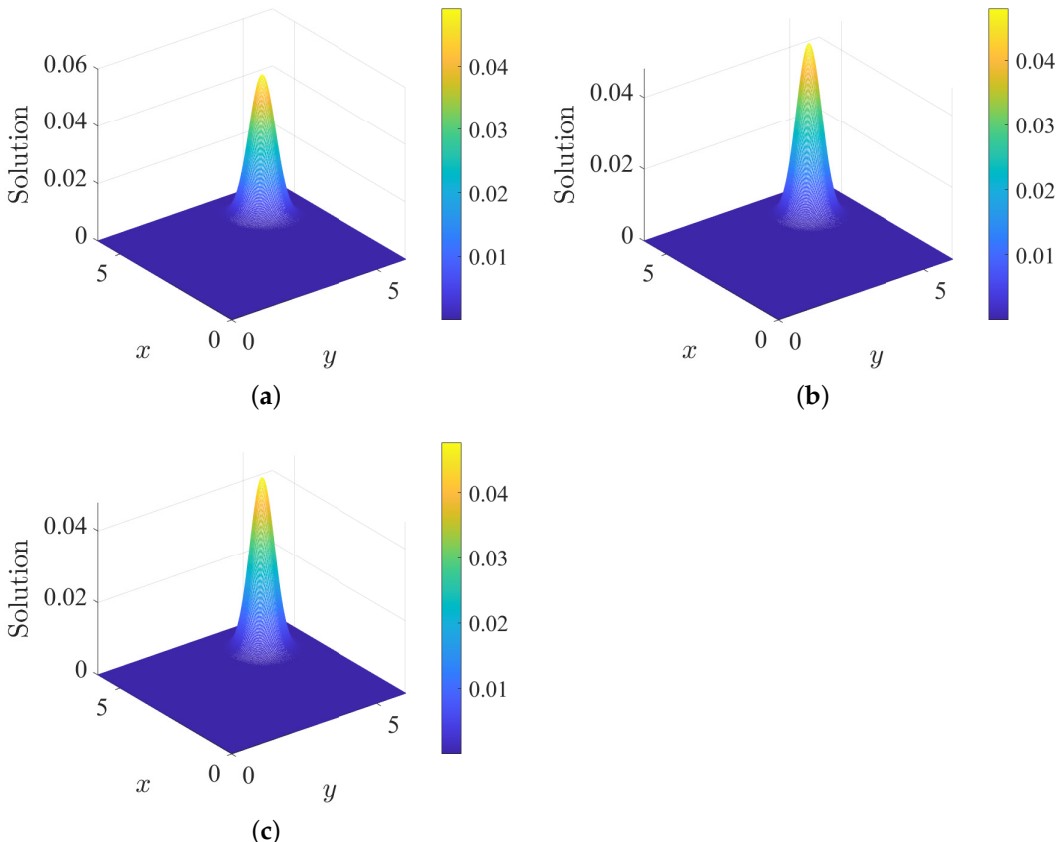

**Figure 9.** The 3D plot of the solution using the Lax–Wendroff scheme vs. $x$ vs. $y$ at time $T = 5$ with $h = 0.025$, $U = V = 0.8$, $K_x = K_y = 0.01$, $x, y \in [0, 6] \times [0, 6]$. (**a**) $k = 0.00625$; (**b**) $k = 0.0040$; (**c**) $k = 0.0025$.

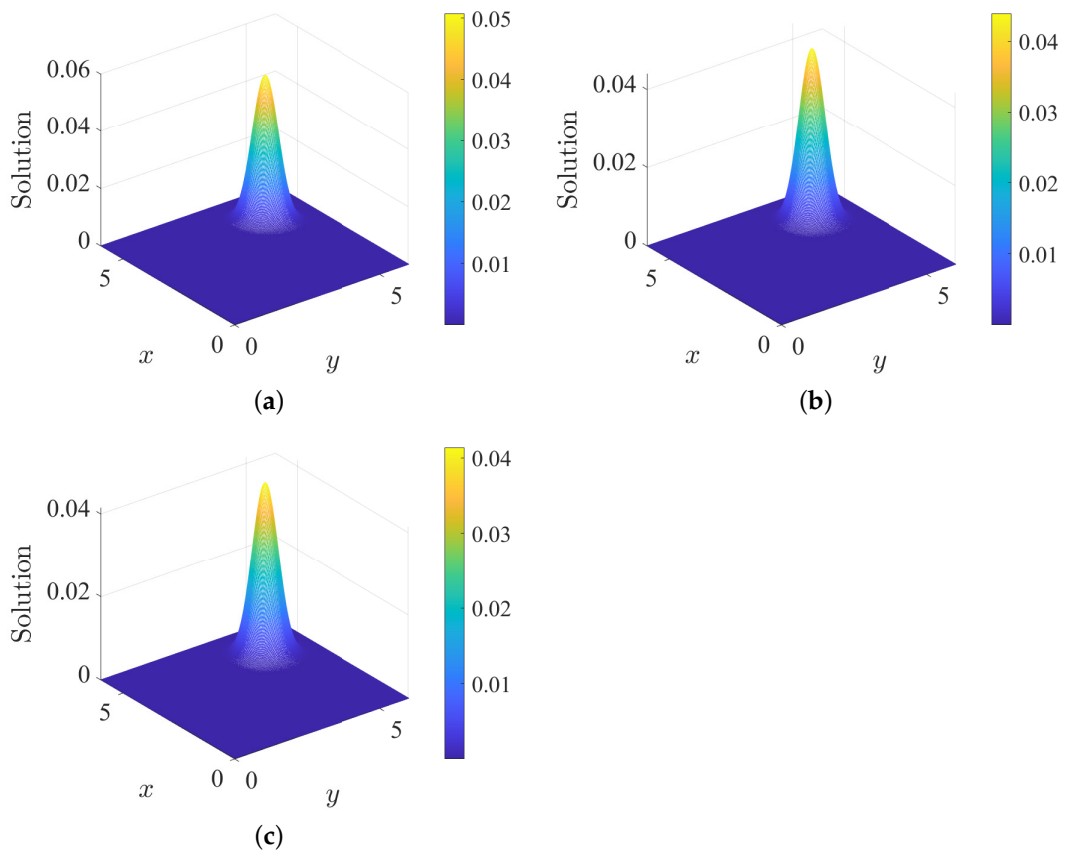

(a)

(b)

(c)

**Figure 10.** The 3D plot of the solution using the NSFD scheme vs. $x$ vs. $y$ at time $T = 5$ with $h = 0.025$, $U = V = 0.8$, $K_x = K_y = 0.01$, $x, y \in [0, 6] \times [0, 6]$. (**a**) $k = 0.00625$; (**b**) $k = 0.0040$; (**c**) $k = 0.0025$.

**Table 5.** $l_2$, $l_\infty$ errors, and $l_2$ rate of convergence in time from the three methods for $U = V = 0.8$, $K_x = K_y = 0.01$ at time $T = 5$ using $h = 0.025$ and different values of $k$.

| Schemes | Value of $k$ | $l_2$ Error | $l_\infty$ Error | Rate of Convergence in Time $(l_2)$ |
|---|---|---|---|---|
| Kowalic–Murty scheme [11] | 0.008 | 0.0085 | 0.0188 | – |
| | 0.004 | 0.0034 | 0.0070 | 1.3219 |
| | 0.002 | 0.0015 | 0.0031 | 1.1806 |
| Lax–Wendroff | 0.008 | 0.0035 | 0.0046 | – |
| | 0.004 | 0.0017 | 0.0026 | 1.0418 |
| | 0.002 | $9.7500 \times 10^{-4}$ | 0.0018 | 0.8021 |
| NSFD | 0.008 | 0.0038 | 0.0049 | – |
| | 0.004 | 0.0022 | 0.0039 | 0.7885 |

### 8.2. Numerical Experiment 4

Here, we present the numerical results for the case involving the two-dimensional convection–diffusion equation for a point source. The concentration $C$ at a given time $t$ for a point source pollutant placed at the center of the given domain $(x_0, y_0)$ is given and the transport of the instantaneous point source pollutant originally placed at the center of the domain is obtained using the three schemes. We observe that the initial Gaussian pulse located at $(x_0, y_0) = (50,000, 50,000)$ spread through the entire domain. Figures 11–13 show the transport of the instantaneous point source pollutant originally placed at the center of the domain for different values of temporal step sizes, for the three schemes, respectively.

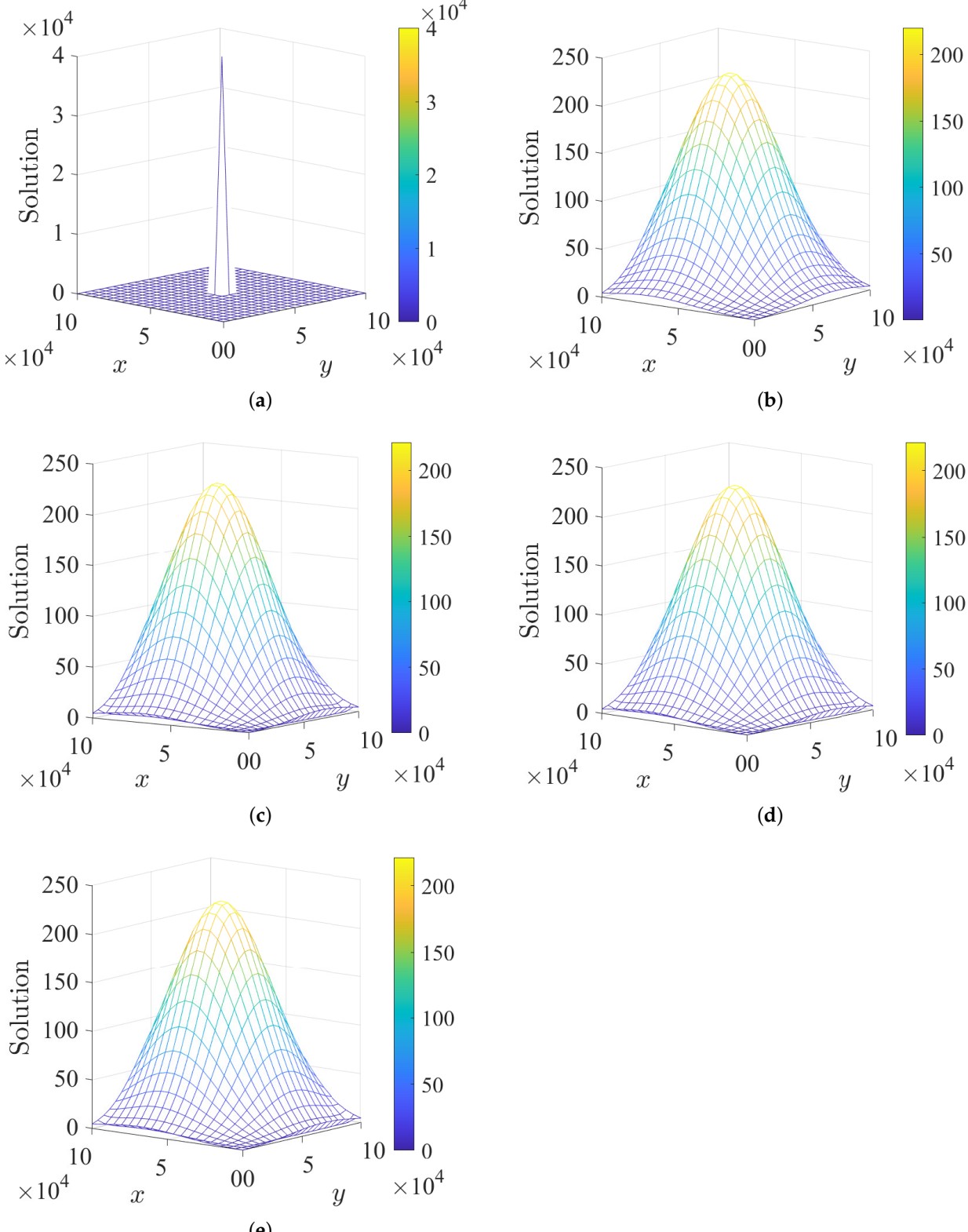

**Figure 11.** Plot of numerical solution vs. $x$ vs. $y$ at time $T = 36,000$ when $h_x = h_y = 5000$, $(x_0, y_0) = (50,000, 50,000)$, $U = V = 0.5$, $K_x = K_y = 10,000$, $K = 10^{12}$ on $[0, 100,000] \times [0, 100,000]$ using the scheme from the Kowalic–Murty scheme. (**a**) Initial profile; (**b**) exact solution; (**c**) numerical profile at $k = 50$; (**d**) numerical profile at $k = 30$; (**e**) numerical profile at $k = 20$.

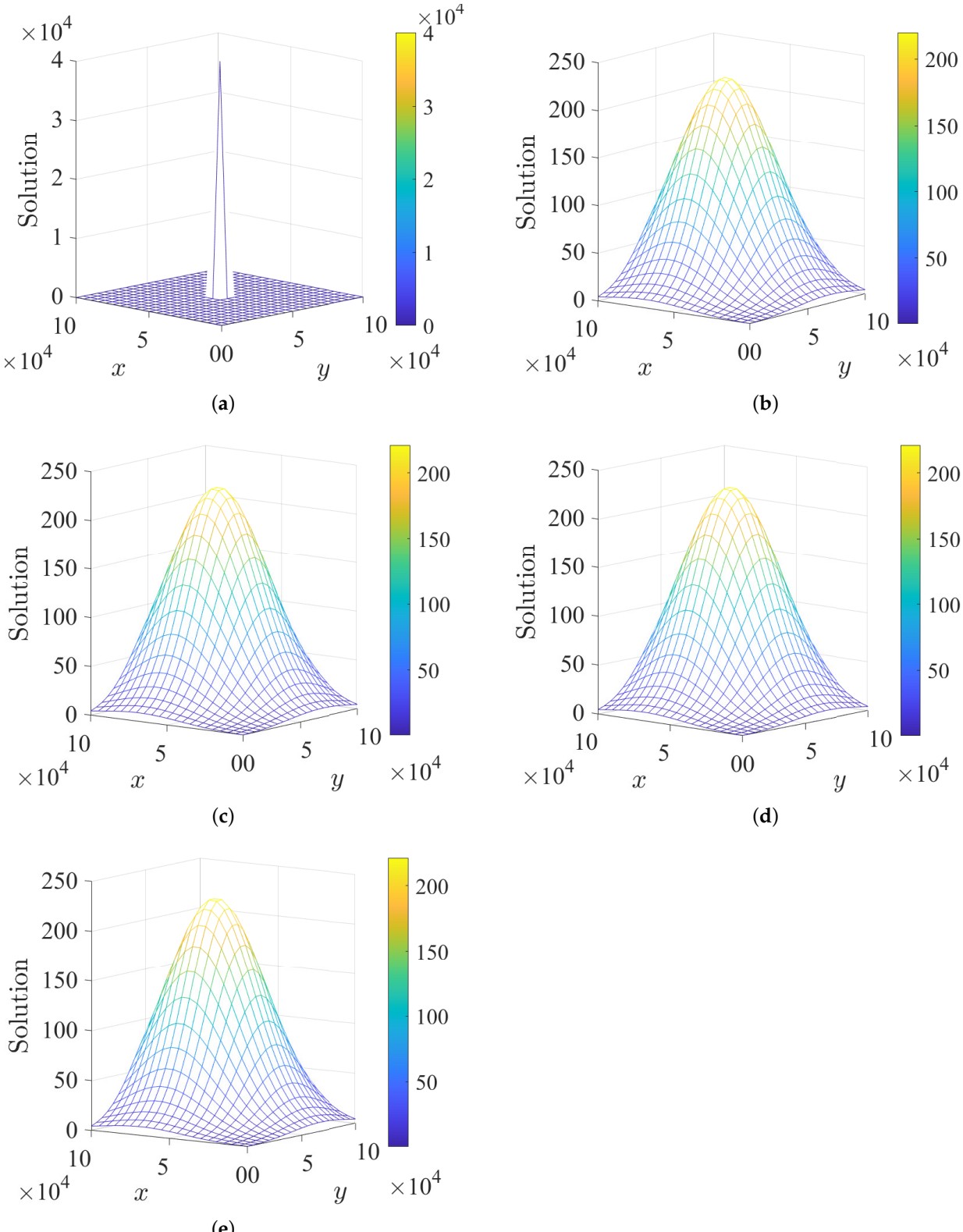

**Figure 12.** Plot of the numerical solution vs. $x$ vs. $y$ at time $T = 36,000$ when $h_x = h_y = 5000$, $(x_0, y_0) = (50,000, 50,000)$, $U = V = 0.5$, $K_x = K_y = 10,000$, $K = 10^{12}$ on $[0, 100,000] \times [0, 100,000]$ using Lax–Wendroff. (**a**) Initial profile; (**b**) exact solution; (**c**) numerical profile at $k = 50$; (**d**) numerical profile at $k = 30$; (**e**) numerical profile at $k = 20$.

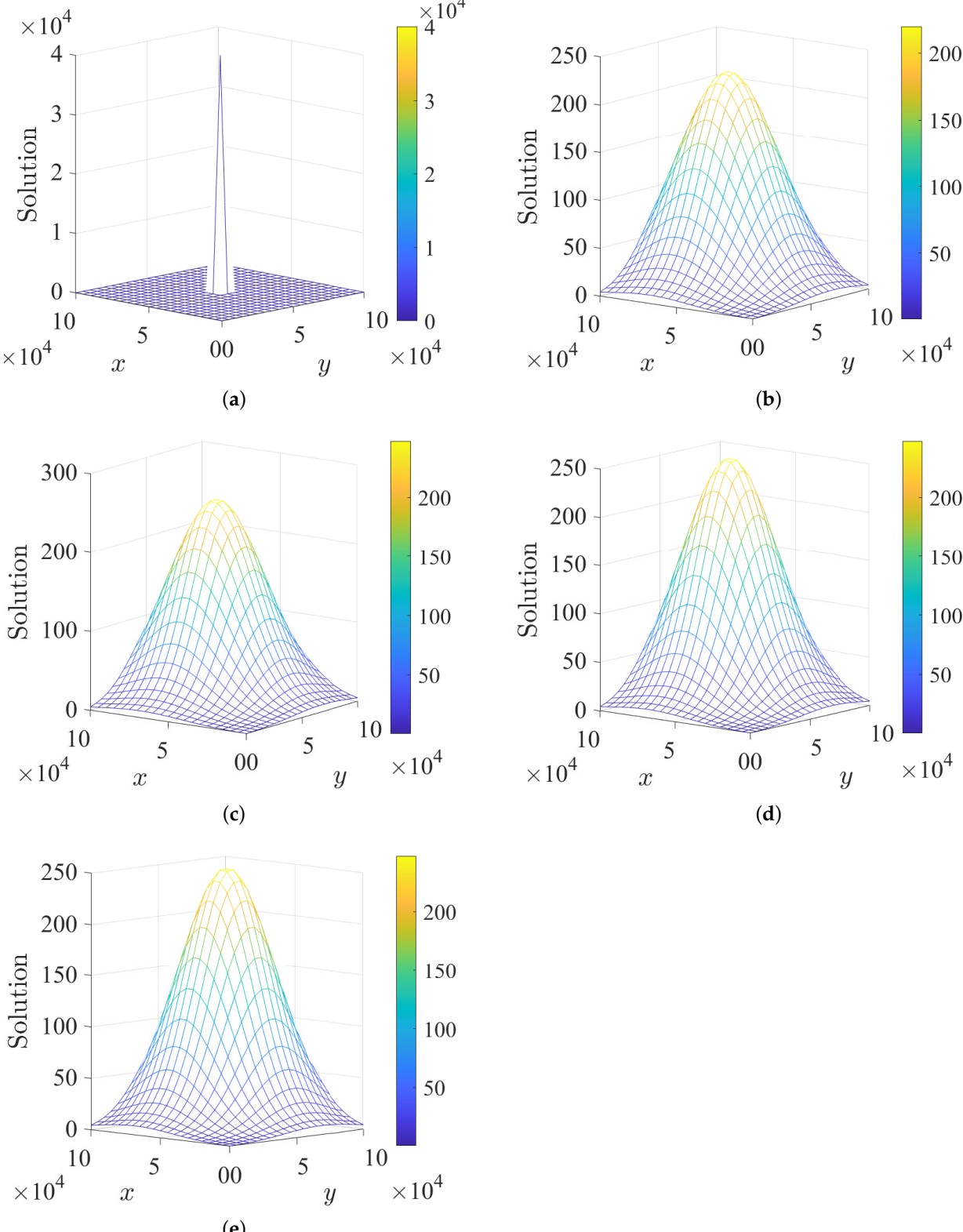

**Figure 13.** Plot of the numerical solution vs. $x$ vs. $y$ at time $T = 36,000$ when $h_x = h_y = 5000$, $(x_0, y_0) = (50,000, 50,000)$, $U = V = 0.5$, $K_x = K_y = 10,000$, $K = 10^{12}$ on $[0, 100,000] \times [0, 100,000]$ using NSFD. (**a**) Initial profile; (**b**) exact solution; (**c**) numerical profile at $k = 50$; (**d**) numerical profile at $k = 30$; (**e**) numerical profile at $k = 20$.

Table 6 shows the errors for the three schemes when $h_x = h_y = 5000$, $k = 50$, $(x_0, y_0) = (50,000, 50,000)$, $U = V = 0.5$, $K_x = K_y = 10,000$, $K = 10^{12}$ at time $T = 36,000$

on $[0, 100, 000] \times [0, 100, 000]$. In this case, it is observed that the $l_2$ and $l_\infty$ errors for the Lax–Wendroff are least followed by the Kowalic–Murty scheme [11]. The NSFD scheme has the largest error in this case, as depicted in Table 6.

**Table 6.** $l_2$ and $l_\infty$ errors at time $T = 36,000$ using the three methods for $U = V = 0.5$, $K_x = K_y = 10,000$, $K = 10^{12}$ at $h_x = h_y = 5000$ and different values of $k$.

| Schemes | Value of $k$ | $l_2$ Error | $l_\infty$ Error |
|---|---|---|---|
| Kowalic–Murty scheme [11] | 300 | $4.3103 \times 10^5$ | 14.0042 |
| | 150 | $4.2544 \times 10^5$ | 14.0042 |
| Lax–Wendroff | 600 | $1.5957 \times 10^5$ | 5.7115 |
| | 300 | $8.2684 \times 10^3$ | 0.3271 |
| | 150 | $4.0194 \times 10^4$ | 1.4263 |
| NSFD | 600 | $8.4336 \times 10^5$ | 26.6466 |
| | 300 | $8.5971 \times 10^5$ | 27.1736 |
| | 150 | $8.6977 \times 10^5$ | 27.5904 |

## 9. Conclusions

We used three numerical schemes to solve one- and two-dimensional convection–diffusion equations where the initial conditions consisted of symmetrical profiles. Four cases were considered with different coefficients of advection and dissipation. The most efficient scheme was the Lax–Wendroff in all four experiments. We obtained better results using Lax–Wendroff as compared to the results obtained by Sankaranarayanan et al. [10] when they used the scheme constructed by Kowalik and Murty [11]. We also computed the optimal time step size by minimizing the dispersion error for one of the cases and this was validated. Convergence tests were carried out and the numerical rate of convergence was found to be close to the theoretical one.

## 10. Future Work

For this current study, we used three finite difference schemes to solve the convection–diffusion equation. Since the modeling of the oil spill has convection and diffusion terms, we will extend this work to solve mathematical models of oil spills in the literature. A good example of the governing equations of 3D tidal flows as well as oil concentration distributions in coastal waters are given in [38].

**Author Contributions:** Conceptualization: A.R.A. and O.A.J.; methodology: O.A.J., H.H.G. and A.R.A.; software: O.A.J. and H.H.G.; validation: O.A.J., H.H.G. and A.R.A.; formal analysis: O.A.J., H.H.G. and A.R.A.; investigation: all authors; resources: O.A.J., H.H.G. and A.R.A.; writing—original draft preparation: O.A.J., H.H.G. and A.R.A.; typing: H.H.G. and O.A.J.; writing—review and editing: all authors; supervision: A.R.A.; project admin: A.R.A.; funding acquisition: A.R.A. All authors have read and agreed to the published version of the manuscript.

**Funding:** O. A. Jejeniwa was funded by the Nelson Mandela University Council Postdoc Fellowship from March 2021 to September 2022 to carry out this research.

**Data Availability Statement:** Not applicable.

**Acknowledgments:** A. R. Appadu is grateful for the funding from the Nelson Mandela University, which enabled him to carry out this work. The authors are grateful to the three anonymous reviewers who provided significant feedback, which considerably improved the paper.

**Conflicts of Interest:** The authors declare no conflict of interest.

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
