# Peer review of "Numerical Modeling of Pollutant Transport: Results and Optimal Parameters"

_symmetry, doi:10.3390/sym14122616_

Round 1

Reviewer 1 Report

Review Report

Title:

Numerical Modelling of Pollutant Transport: Results and Optimal Parameters

Authors described finite-difference discretization technique and non-standard scheme for solving multi-dimensional convection dominated diffusion equations. The convergence rate and error estimates  (L2, L-infinity) errors corroborate the utility of proposed scheme. A detailed stability analysis and simulations results are presented to justify the new technique.  The paper is well presented and worth for the researchers working in the area of computational partial differential equations and mathematical modelling.

I have some observations:

1) In the abstract:

“The numerical methods used are: Kowalic & Murty…..” should be written as. “The numerical methods used are Kowalic & Murty….”

Replace “Using  Lax-Wendroff scheme, we showed….” by “Using the Lax-Wendroff scheme, we showed…”

2) In Introduction

Replace

“We have been using petroleum as a source of fuel for daily human activities. The use of crude oil however, did not in the actual sense increase until the industrial revolution, where oil became valuable as a fuel for illumination and a lubricant before it became a replacement for coal, animal power, wood and other sources of energy. Liquid petroleum has some useful advantages over other energy source of the rimes; it was concentrated and could be easily transported from one point to another. The transportation of petroleum from one country to another is mostly done through cargo ship and pipelines submerged in marine environment carrying large amount of petroleum across the open and coastal seas [3,4].”

by

“We have been using petroleum as fuel for daily human activities. The use of crude oil, however, did not, in the actual sense, increase until the industrial revolution, when oil became valuable as a fuel for illumination and a lubricant before it became a replacement for coal, animal power, wood, and other sources of energy. Liquid petroleum has some useful advantages over other energy sources of the rimes; it is concentrated and could be easily transported from one point to another. The transportation of petroleum from one country to another is mostly done through cargo ships and pipelines submerged in the marine environment carrying a large amount of petroleum across the open and coastal seas [3,4].”

3) I suggest to carefully check the grammar in the entire manuscript. It is difficult for the reviewer to see the non-mathematical contents with a utmost care.

4) Page 3: Put the index i below max in the definition of l-infinity.

5) Page 10, Section 4.3, Last Line:   This scheme is of order one in space and time.

The first order scheme often fails to achieve numerically stable solution and oscillatory behaviour appears with small diffusion parameter. The estimated results can be improved by employing high-order scheme. I suggest authors to incorporate following references

https://doi.org/10.37256/rrcs.1120221466

https://doi.org/10.1002/num.22702

https://doi.org/10.3390/sym13071123

https://doi.org/10.1108/HFF-01-2016-0038

and mention in the introduction that  “High-order scheme for convection-diffusion equations are reported in the past by ……”.

6) Section 4.1:  Kowalic & Murty (1993) scheme

del C/ del x is approximated using four points, while del^2 C/ delx x^2 is approximated with three-points. Is it possible to  discretize  del C/ del x with three-point or two-point only. By considering more points, does it affect stability and convergence order?

The appearance of C(i-2) may require fictitious points, Is it ?

The above observation is resolved in Eq. (15) by Lax-Wendroff.

The consistency in 4.1 and 4.2 are nicely presented.

7) What is the effect of frequency parameter U in the nonstandard scheme (see Eq. (26))?

The numerical simulations and graphical illustrations are well presented.

With these modifications, I suggest the paper may be accepted for publication.

Author Response

The authors are grateful to reviewer 1 for the comments and suggestions.  The authors revised the paper.  Responses to reviewer 1 are available in enclosed pdf.

Reviewer 2 Report

This paper presented results using three numerical schemes for solving one and two dimensional convection diffusion equations. It is used for the modelling of coastal environment with initial conditions being symmetric profiles. The accuracy of this schemes were tested by comparing results with known analytical solutions. Some useful results are obtained. But it exists some problems which mentioned as following,

1. The novel points of this paper are not clear. The numerical schemes and the numerical results are not new for this topic.

2. As some numerical schemes are not new coming, it should omit some basic theories and equations. It should highlight the novel point of the improvement.

3. The setting of the numerical model should be given more clear. The size of the model, the mesh, the boundary conditions and so on.

4. For the numerical results, it should give the improvements of the proposed  method, velocity or pollution distribution over the whole calculation domain.

5. The conclusions should be rewritten, it should introduce the significant improvement according to the numerical results and comparisons.

According to the problems as mentioned above, this paper should be revised carefully.  

Author Response

The authors are grateful to comments from reviewer 2 which enabled them to improve quality and presentation of the paper.  Responses to reviewer 2 are enclosed in pdf.

Reviewer 3 Report

This paper considers numerical scheme for 1D & 2D convective diffusion equation. First, for the set of equation considered, there is no challenge in numerical simulation any more. Second, based on the simulations, there is no new insight added to the field. Third, the paper considers 1D and 2D cases, but Equations 1-4 shows 3D equations. Fourth, why do not use Equation 4 for all boundaries? Fifth, above Equation 5, this is not "perturbation". Based on these, I would suggest to reject its publication.

Author Response

The authors are grateful to feedback from reviewer 3 which enabled them to considerably improve the paper.  Reponses to reviewer 3 are available in enclosed pdf. 

Round 2

Reviewer 2 Report

This paper has not been revised according to the referee's comment, it should give the responses according to the referee's comments one by one. This revised paper can not be acceptable.

Author Response

The authors are grateful to comments raised by reviewer 2.  All the requested corrections were made.  Kindly see the document enclosed (Response to reviewer 2.pdf)

Round 3

Reviewer 2 Report

The authors have tried them best to revise the paper, it should adjust the arrangement of some charts to make the paper more easy to read. It can be accepted after minor revision.

Author Response

The paper has been read many times and some sentences removed to decrease the similarity index from 16 %.  The results are better presented; initially some legends were hiding the graphs and this matter has been resolved.  Also, orientation of the figures is improved. The introduction and conclusion have been checked properly.